# Platelet Biochemistry and Morphology after Cryopreservation

**DOI:** 10.3390/ijms21030935

**Published:** 2020-01-31

**Authors:** Katrijn R. Six, Veerle Compernolle, Hendrik B. Feys

**Affiliations:** 1Transfusion Research Center, Belgian Red Cross Flanders, 9000 Ghent, Belgium; 2Faculty of Medicine and Health Sciences, Ghent University, 9000 Ghent, Belgium; 3Blood Service, Belgian Red Cross Flanders, 2800 Mechelen, Belgium

**Keywords:** platelets, transfusion, storage, cryopreservation

## Abstract

Platelet cryopreservation has been investigated for several decades as an alternative to room temperature storage of platelet concentrates. The use of dimethylsulfoxide as a cryoprotectant has improved platelet storage and cryopreserved concentrates can be kept at −80 °C for two years. Cryopreserved platelets can serve as emergency backup to support stock crises or to disburden difficult logistic areas like rural or military regions. Cryopreservation significantly influences platelet morphology, decreases platelet activation and severely abrogates platelet aggregation. Recent data indicate that cryopreserved platelets have a procoagulant phenotype because thrombin and fibrin formation kicks in earlier compared to room temperature stored platelets. This happens both in static and hydrodynamic conditions. In a clinical setting, low 1-h post transfusion recoveries of cryopreserved platelets represent fast clearance from circulation which may be explained by changes to the platelet GPIbα receptor. Cryopreservation splits the concentrate in two platelet subpopulations depending on GPIbα expression levels. Further research is needed to unravel its physiological importance. Proving clinical efficacy of cryopreserved platelets is difficult because of the heterogeneity of indications and the ambiguity of outcome measures. The procoagulant character of cryopreserved platelets has increased interest for use in trauma stressing the need for double-blinded randomized clinical trials in actively bleeding patients.

## 1. Introduction

Transfusion of platelet concentrates (PC) is a life-saving intervention for patients suffering from acute blood loss but is more often used as a supportive prophylactic therapy for patients with diverse hematologic diseases. The demand for PC has increased over the past years. This is caused by a growing number of elderly patients who suffer from (chronic) hematologic malignancies like myeloid leukemias and non-Hodgkin lymphomas [1].

PC are stored at room temperature (RT, 22 °C) with constant agitation. This storage temperature was shown to yield superior platelet recoveries after transfusion compared to cold storage (4 °C) [2]. However, RT storage severely limits PC shelf-life to maximally 7 days. The exact shelf-life depends on the regulation by country. The short shelf-life is a precautionary measure that minimizes the chance for bacterial bloom. In addition, ex vivo platelet storage induces a series of biochemical and functional changes to the platelet, collectively called platelet storage lesion [3]. This phenomenon causes gradual platelet dysfunction and eventually cell death rendering the PC of poor quality after longer term RT storage. Efficient PC stock management is therefore highly challenging and this is bound to worsen by increasing demands, increasing operational activities (like pathogen inactivation) [4] and/or increasing tests for transmittable diseases. Although challenging, PC banking is still manageable in highly structured, densely populated and well-organized regions but becomes increasingly problematic when logistics are hampered by vast distances, harsh terrain, lack of facilities and/or warfare.

Alternatives for RT storage have been investigated since the 1970s mainly to improve PC supply to the battlefield. Both cold storage (4 °C) and cryopreservation (−96 °C to −180 °C) have been studied. The current review will focus on platelet cryopreservation. For reviews on cold storage we refer to published work [5,6,7]. In the mid-seventies, Valeri et al. developed a cryopreservation protocol using 6% (*v*/*v*) dimethylsulfoxide (DMSO) as cryoprotectant and storage at −80 °C without controlled-rate freezing [8]. The in vivo platelet recoveries and platelet survival were acceptable in healthy volunteers with aspirin-induced thrombocytopathy [8]. To limit toxicity, DMSO had to be removed by centrifugation and washing before platelet transfusion. These additional manipulations resulted in significant practical limitations of the protocol, especially for use in the military where fast supply and thus minimal product manipulation is required. Research into cryopreservation continued but focused on its potential use in patientsrefractory to platelet transfusions instead of for military purposes.

The publication of a no-wash protocol by Valeri et al. in 2005 presented a gold standard cryopreservation protocol [9,10] and boosted the research field. The method hyperconcentrates the platelets by centrifugation after the addition of 6% (*v*/*v*) DMSO. This way the platelet-free supernatant can be removed prior to freezing resulting in the effective removal of a large fraction of DMSO beforehand. After cryopreservation, the hyperconcentrated platelet pellet is reconstituted in either plasma or saline. Since this landmark publication, interest in cryopreservation is renewed especially in the military or in vast rural areas although the technique may equally function to support regular blood banks to further reduce risks of stock failure [10].

Despite substantial research into platelet cryopreservation, lots of questions remain. This review will summarize the biochemical and molecular changes in platelets following cryopreservation and will link these data to results obtained in clinical trials. In doing so we wish to reveal the gaps in the research and stimulate further development in this field.

## 2. Biochemical Changes to Platelets Following Cryopreservation

### 2.1. At the Level of Single Platelets

#### 2.1.1. Platelet Viability and Recovery

Viable nucleated cells bear an intact nucleus with cellular chromatin. The chromatin is not detectable using live/dead staining with dyes such as propidium iodide, 7-aminoactinomycin D (7-AAD) or trypan blue. Loss of membrane integrity in dying or dead cells allows entry of the dye inside the cell and this stains the chromatin and marks the cell as non-viable. Platelets are anucleate and lack chromatin so they do not divide nor differentiate. Therefore, viability staining in platelets is questionable. As an alternative to chromatin live/dead staining acetoxymethylesters of calcein and related fluorophores can be used to selectively label platelets that contain active esterases [11]. It remains however to be demonstrated what a positive calcein stain then actually means and whether this “viable” tag implies that the platelet is functional. Other methods that have assessed “viability” in the context of cryopreservation research are extracellular lactate dehydrogenase activity [12], hypotonic stress response [13,14] and mitochondrial membrane potential [11,15] but there is no evidence that these are more appropriate than calcein staining for the “viability” outcome parameter.

An alternative parameter often tested in cryopreservation is platelet recovery. This represents the number of platelets in the product after cryopreservation relative to that before. Because platelet morphology is significantly affected by cryopreservation, recovery is dependent on how a platelet is defined. A more appropriate definition therefore is the number of recovered particles with a size and shape similar to platelets relative to the number of platelets before. Recovery is largely influenced by platelet losses during handling as part of the cryopreservation and reconstitution method. Our data demonstrate that approximately 12.5% (±2.1) (mean ± SD, *n* = 12) of the platelets are lost in the supernatant during the hyperconcentration step of the no-wash protocol published by Valeri et al. [10]. Consequently, caution is needed when interpreting published in vitro platelet recoveries because these reflect the method’s efficiency and not platelet quality.

#### 2.1.2. Platelet Morphology

Platelet morphology is significantly affected during cryopreservation. Most studies indicate an increase in mean platelet volume measured by a hematology analyzer [16,17]. Electron microscopy images, however, do not suggest an actual increase in platelet size but clearly demonstrate changes in platelet shape instead. RT stored platelets have a typical disc shape while cryopreserved platelets appear more spherical or balloon-shaped [18,19].

Many cryopreserved platelets moreover have morphologic features reminiscent of activated platelets. This includes an irregular ruffled cell surface and an increased number of pseudopodia (Figure 1). A minority of cryopreserved platelets displays a condensed morphology marked by cytoplasmic membrane disintegration [12,14,18,19]. These shape changes reduce the archetypical anisotropic morphology of RT stored platelets that results in the strong diffraction of light and that is often used as a quality outcome parameter in blood banks called “swirl” [20]. Consequently, cryopreserved platelets have significantly less or even no swirl compared to RT stored platelets [21].

#### 2.1.3. Changes to the Cytoplasmic Membrane

Cryopreservation causes a significant increase in extracellular vesicle (EV) content (synonyms: microparticles, microvesicles or exosomes) in PC. These EV have variable diameters ranging from 20 to 200 nm. The EV population is characterized by the expression of high levels of the aminophospholipids phosphatidylethanolamine (PE) and -serine (PS) [16,19]. Exact enumeration of EV in PC is difficult because of the detection limit on particle size in typical flow cytometers and because definitions of platelet EV are ambiguous. However, in general, EV content increases approximately fivefold after cryopreservation [14,17,18,19].

PS/PE expression is not limited to the EV population but applies to normally sized cryopreserved platelets as well. In resting RT stored platelets, PS/PE is actively kept on the inner leaflet of the cytoplasmic membrane (Figure 1). The aminophospholipids however flop to the outer membrane during platelet activation and/or platelet apoptosis. This loss of membrane asymmetry can be detected in a Ca^2+^-dependent manner by fluorescently labeled Annexin V or lactadherin using flow cytometry [16,17,19,22].

Next to loss of membrane asymmetry, membrane integrity of platelets is also affected during cryopreservation. In an experiment using platelets that were preloaded with fluorescein, 40% of the cytoplasmic dye was released in the supernatant following cryopreservation with 5% (*v*/*v*) DMSO [23]. This was twofold less than in conditions without cryoprotectant but nonetheless indicates significant damage to membrane integrity. In addition, the loss of platelet molecules by freezing is substantially addressed in studies on human platelet lysate which is prepared exactly by (repeated) freeze-thawing [24,25]. It therefore seems worthwhile investigating if other molecules are lost from cytoplasma during cryopreservation. Of note, the release of cytoplasmic molecules should not be confused with granule release which is a form of controlled exocytosis and is not related to membrane damage.

#### 2.1.4. Surface Receptor Expression

P-selectin (CD62P) is a receptor residing inside the resting platelet on membranes of α-granules. During platelet activation, the α-granules fuse with the cytoplasmic membrane releasing its content in the surrounding medium in a process called degranulation. This causes integration of P-selectin in the cytoplasmic membrane and exposes the receptor to the platelet exterior (Figure 1). Cryopreservation results in a significant increase of P-selectin expression on the platelet surface implying that cryopreservation causes spontaneous degranulation [11,12,18,19,23,26,27]. This may not necessarily be the same as platelet activation in this case because integrin α_IIb_β_3_ activation is not substantially increased after cryopreservation. Using PAC-1 in flow cytometry to specifically label platelets that express activated integrin α_IIb_β_3,_ we and others only found a small increase in the fraction of platelets expressing activated integrin and no difference in PAC-1 fluorescence after cryopreservation [12,15,17,26,28,29].

Another platelet specific receptor is the GPIbα portion of the GPIbα/IX/V complex that binds von Willebrand factor (VWF) for platelet adhesion during primary hemostasis. Surface receptor expression of GPIbα on cryopreserved platelets has been studied extensively over the years with some important observations. A significant decrease in GPIbα receptor expression is consistently found on cryopreserved platelets using flow cytometry [12,15,17,26,29,30]. This can be explained by ectodomain shedding of GPIbα resulting in the release of the extracellular soluble part of the receptor called glycocalicin. This glycosylated molecule can be found at low levels in the supernatant of regular PC during RT storage. It is also consistently found at low concentrations in plasma from healthy donors. Glycocalicin levels are significantly increased in the supernatant of cryopreserved platelets [11,17] implying that sheddases are activated by cryopreservation.

Cold storage (4 °C) of platelets may cause GPIbα receptor clustering and deglycosylation of N-glycans, a phenomenon that leads β-N-acetylglucosamine (β-GlcNAc) exposure and accelerated clearance from circulation [31,32,33]. These characteristic changes to GPIbα hydrocarbon sidechains are however not necessarily applicable to cryopreserved platelets because Waters et al. could not find significant GPIbα clustering and Zhao et al. found no increased exposure of β-GlcNAc after cryopreservation [28,34].

When measuring CD42b in cryopreserved platelets using flow cytometry two subpopulations can often be distinguished based on fluorescence. One subpopulation has a normal CD42b signal and thus normal GPIbα expression and the other has a significantly lowered signal. This phenomenon was observed both in human and in baboon cryopreserved platelets. In general, approximately half of cryopreserved platelets have low GPIbα expression (GPIbα^low^) while the other half has normal GPIbα expression (GPIbα^normal^) [11,12,17,26,30]. Specific features and relevance of this phenomenon will be discussed in detail in paragraph 3.

#### 2.1.5. Metabolic Changes

Both mitochondrial membrane potential and hypotonic shock response were used as determinants of platelet metabolism. Both parameters were significantly affected after cryopreservation suggesting changes to mitochondria and/or anaerobic respiration. How these metabolic changes relate to decreased platelet function or (pro)coagulation and thus thrombus formation is not (yet) known [11,14,15,18].

pH measurements immediately post-thaw are not relevant as such because they solely and entirely dependent on the composition of the solution used for PC reconstitution. At that time point platelets do not contribute to pH. It can however be used as a baseline to identify pH changes as a function of subsequent storage post-thaw. Longitudinal follow-up of pH is typically performed by blood institutions as a quality control for RT stored platelet metabolism. It even serves as a gold standard [20]. Because cryopreserved platelets are generally transfused within 4 h after thawing, determination of pH during storage is less relevant and studies provide data limited to a 24 h post-thaw period [9]. The pH of cryopreserved PC did not change in these 24 h post-thaw despite small decreases in glucose and increases in lactate content [11,12,14,28]. Direct comparison of glucose consumption and lactate production of cryopreserved and RT stored platelets did indicate an acceleration of the glycolytic pathway [14]. This could be a result of temperature cycling stress.

#### 2.1.6. Signal Transduction

Only one study investigated changes in signal transduction in vitro after cryopreservation of platelets. In general, a lower degree of phosphorylation was detected on all investigated proteins including Lyn, ERK and Akt after cryopreservation [22]. Phosphorylation of signaling proteins can both positively and negatively regulate platelet function and so, an overall lower phosphorylation activity may explain certain observations. Unraveling signal transduction pathways after cryopreservation is therefore required to better understand what signaling cues are affected in platelets during cryopreservation.

### 2.2. Platelet Function

#### 2.2.1. Agonist-Induced Integrin Activation and Aggregation

Platelet aggregation is still considered the clinical gold standard for detecting platelet defects [35]. Therefore, when aggregation is significantly, affected platelets are often considered overall dysfunctional. Cryopreserved platelets respond poorly in light transmission aggregation when single agonists are used at concentrations that typically cause maximal aggregation in RT stored platelets. Cryopreserved platelets will generally not respond to weak agonists like adenosine diphosphate (ADP) but this is often the case for RT stored platelets as well [36].

However, aggregation is detected with strong agonists like thrombin or collagen added at high concentrations [22]. Furthermore, when several agonists are combined like thrombin receptor agonist peptide 6 (TRAP-6) with ADP and epinephrine substantial aggregation is detected in cryopreserved platelets albeit two- to three-fold lower than in RT stored platelets [17,37]. Ristocetin induced agglutination requires less outside-in signaling to crosslink platelets via GPIbα and VWF compared to agonist induced aggregation which crosslinks platelets via fibrinogen and integrin α_IIb_β_3_. Therefore, agglutination of cryopreserved platelets is possible using 1.5 mg/mL ristocetin. The amplitude is about 75% that of RT stored platelets [17].

These findings are in line with measurements of integrin α_IIb_β_3_ activation using PAC-1 in flow cytometry. This outcome measure is significantly decreased after cryopreservation in response to single agonists like ADP, collagen, TRAP-6, epinephrine, thrombin or cross-linked collagen-related peptides (CRP-XL). It should be noted that the magnitude of the response differs depending on the nature of the agonist but decreases between cryopreserved and RT stored controls were apparent in both buffy coat and apheresis-derived PC [12,15,17,22,26]. Both the aggregation and integrin α_IIb_β_3_ activation assays indicate the requirement for strong platelet stimulation after cryopreservation to obtain measurable amplitudes of platelet activation. However, even at high concentrations of strong agonists the activation level of cryopreserved platelets is lower than the responses obtained by RT stored platelets using the same test conditions.

#### 2.2.2. Platelet Adhesion and Coagulation in Hydrodynamic Flow

Adhesion to extracellular matrix proteins (ECM) of cryopreserved platelets in reconstituted blood was investigated by two groups using different models, i.e., the Baumgartner model and a real-time microfluidic flow chamber model, respectively [38,39,40]. Using the Baumgartner model, Cid et al. found 10% surface coverage on denuded rabbit aorta using cryopreserved platelets perfused at a wall shear rate of 600 s^−1^, while surface coverage was 23% for RT stored platelets [41]. We investigated platelet adhesion by perfusion of reconstituted whole blood onto collagen only or onto collagen and tissue factor (TF) in microfluidic flow chambers at a wall shear rate of 1000 s^−1^. The adhesion rate of cryopreserved platelets was decreased twofold compared to paired platelets before cryopreservation [17]. Despite differences in technical set-up, wall shear rate and experimental endpoint, both independent experiments show significant impaired platelet adhesion after cryopreservation.

Both models were also used to assess coagulation in the presence of cryopreserved platelets. Fibrin coverage after 4 min of perfusion in the Baumgartner model was not significantly different between RT stored and cryopreserved platelets despite the lower number of adhered platelets [41]. In our microfluidic system fibrin formation onset was even faster in the presence of cryopreserved platelets compared to control non-cryopreserved cells [17]. This implies that adherent cryopreserved platelets are more procoagulant than regular platelets because despite the twofold decreased platelet coverage, fibrin formation is at least as quick or even quicker.

#### 2.2.3. Coagulation in Static Conditions

Coagulation is mostly studied in static conditions, i.e., in the absence of hydrodynamic flow. Dynamic measurement of thrombin generation is performed by thrombin generation assays (TGA) and fibrin formation can be studied using rotational thromboelastometry (ROTEM). These assays can provide information on the (pro)coagulant properties of (cryopreserved) platelets because platelets are natural catalysts of coagulation.

Results from TGA across labs should however be compared with caution. Variables like platelet count, anticoagulant concentration, buffer composition and TF concentration are not standardized and can have a significant impact on the absolute values obtained by TGA [42]. To make the TGA dependent on platelets, a low TF concentration should be used and all other variables should be kept constant. When performed as such, cryopreserved and RT stored platelets can both catalyze thrombin generation in a platelet dose-dependent manner. However, peak thrombin concentrations are twofold higher in the presence of cryopreserved platelets compared to RT stored platelets. Furthermore, lag times are significantly shorter in the presence of cryopreserved platelets compared to RT stored platelets [17,19].

ROTEM is a standardized clinical test that provides dynamic information on the speed of coagulation initiation, propagation and on clot strength expressed respectively as clotting time (R-time), clot formation time (K-time) and maximal amplitude. This assay is particularly dependent on fibrinogen concentrations but if fibrinogen levels are kept constant the subtle contribution of platelets can be measured as well. Tests in ROTEM confirmed the above mentioned findings in TGA including faster clotting time for cryopreserved platelets compared to RT stored platelets. A slight decreased maximal amplitude suggests that the clot is less firm in the presence of cryopreserved platelets compared to RT stored platelets. Contradictory results are published for clot formation time as some groups found a decrease in this parameter after cryopreservation while others found no differences between storage conditions. This discrepancy may be explained by the agonist used to perform ROTEM, e.g., kaolin, ellagic acid or TF [12,16,17].

In conclusion, the data for coagulation obtained by using hydrodynamic flow, TGA and ROTEM all suggest a faster onset of coagulation with higher thrombin production in the presence of cryopreserved platelets compared to RT stored platelets. Whether fibrin clot strength differs between conditions needs to be investigated in more detail.

#### 2.2.4. Procoagulant Cryopreserved Platelets

How can cryopreserved platelets with aberrant morphology, altered surface receptor expression levels and decreased agonist responses be able to promote coagulation similar to or even better than RT stored platelets? Recently, the text book coagulation cascade has been challenged by a novel concept called ‘cell-based coagulation’ [43]. The idea is based on two cell types which are physically separated in healthy tissue, i.e., TF-bearing cells in non-vascular compartments and circulating platelets devoid of TF. In this model, cell-based coagulation is divided in three separate but overlapping phases called initiation, amplification and propagation. Upon injury the leaking blood contacts the interstitial TF-bearing cells leading to release of factor IXa and the formation of the prothrombinase complex (factor Va/Xa) with subsequent generation of small amounts of thrombin. During this initiation phase, platelets adhere to the site of injury onto von Willebrand factor and collagen. Amplification follows when adherent platelets sense the small amounts of thrombin via G-protein coupled protease-activated receptors (PAR) that induce a positive feedback loop in the platelet to catalyze factor V, VIII and XI activation on the platelet surface. In the propagation phase platelets are strongly activated and their increased surface area provides an assembly stage to scaffold prothrombinase and tenase (factor IXa/VIIIa) complexes generating large amounts of (local) thrombin [43,44] (Figure 1).

Cryopreserved platelets possess a pre-activated surface membrane because of the abnormally high aminophospholipid (PS and PE) expression, even in the absence of TF-bearing cells or adhesion to injured vessel wall proteins. Possibly, PS/PE exposure allows skipping the initiation steps of the described cell-based coagulation model. It is known that PS/PE is required to successfully assemble the prothrombinase and tenase complexes on platelets. This may explain the apparent procoagulant character of cryopreserved platelets. The increased binding of coagulation factors to cryopreserved platelets was observed in 1999 by Barnard et al. who demonstrated a 4-fold increased binding of factor V compared to RT stored controls [26]. In addition, the contribution of PS was demonstrated by Tegegn et al. who showed that thrombin generation in the presence of cryopreserved platelets could be significantly decreased in the presence of lactadherin which inhibits assembly of coagulation complexes by blocking PS [19].

Furthermore, cryopreservation increases the EV content significantly. These EV provide additional surface area for coagulation factor/complex adhesion. In addition, platelet-derived EV are 50- to 100-fold more procoagulant than activated platelets alone [45]. This way EV formation during cryopreservation provides not only an increase in surface area but the actual cellular particles may be biochemically better suited to catalyze coagulation. This is demonstrated in an elegant experiment showing that thrombin can still be generated using platelet-poor, EV-rich supernatant of cryopreserved PC albeit at a slower rate than in the presence of cryopreserved platelets [19]. High speed centrifugation of the cryopreserved PC to remove both platelets and EV however resulted in very little and slow thrombin generation. This suggests that maximal procoagulant effects of cryopreservation contain a contribution of both platelets and EV [16,19,29].

The impact of cryopreservation on platelet morphology, biochemistry and function is summarized in Table 1 in comparison to control RT stored platelets.

### 2.3. Clinical Relevance

It is beyond the scope of this review to present a comprehensive overview of all clinical trials performed with cryopreserved platelets in the past because excellent work has been published by others [46]. We would, however, like to highlight the main findings of recent clinical trials and highlight some aspects that still require further investigation.

#### 2.3.1. Recovery/Count Increment and Survival Time

Typical outcome parameters for evaluating transfusion success are platelet recovery and (corrected) count increment ((C)CI). Platelet recovery is mainly determined in test populations of healthy volunteers [8,10,18] while CI is used in patients and is defined as the change in platelet count 1 or 24 h post-transfusion [27,47,48]. This outcome can be corrected for body surface area, i.e., CCI. In most studies cryopreserved platelets are cleared from circulation faster than RT stored platelets. In healthy volunteers, a 50% lower recovery is found when compared to transfusion of RT stored platelets [10,18]. Furthermore, the (C)CI of cryopreserved platelets is significantly lower in patients with hematologic malignancies or in actively bleeding patients when compared to RT stored platelets [27,47,48]. The reason for accelerated clearance is not entirely clear, but changes to platelet receptors may mark the platelet for clearance as is found for cold stored platelets (4°C) in animal models [31,32,49]. Platelet survival was determined in several clinical studies investigating the maximal circulation time of platelets post-transfusion. These data were summarized by Slichter et al. and mean survival time was 7.4 (± 1.4) days for cryopreserved platelets compared to 8.4 (±0.4) days for RT stored platelets which was a significant decrease but still within FDA criteria [46].

#### 2.3.2. Efficacy

Assessing efficacy of (cryopreserved) transfused platelets is not easy because of several reasons. One is the heterogeneity of indications for platelet transfusion ranging from accidental trauma to major surgery to hemato-oncologic and inherited or acquired platelet disorders. For each indication, uniformity of the patient population is difficult to achieve and not all patients present with (serious) bleeding. To sufficiently power clinical studies on bleeding, a large number of patients needs to be included. Furthermore, standardized toolboxes are lacking to objectively measure the platelet contribution in transfusion efficacy. Instead, outcome measures like blood loss, number of blood product transfusions, time between PC transfusion, hemorrhage and cessation of bleeding are most commonly used to determine transfusion efficacy. In addition, although cryopreservation protocols have become increasingly uniform significant variation remains in DMSO concentration, storage temperature (−80 °C vs. −150 °C), freezing rate, resuspension solution (plasma vs. saline) and the type of cryopreservation bag used. Foremost, the platelet concentrate itself varies between studies because platelet separation methods are different throughout the globe. Resulting variables are platelet concentration, platelet additive solution, type of storage bag and pathogen inactivation [37].

Because cryopreserved platelets are procoagulant, they seem ideally suited to treat actively bleeding patients like in (acute) trauma [33,50,51,52]. Clinical trials in acute trauma are a practical challenge and most published studies are observational [53,54]. This renders a low quality score, e.g., by GRADE evaluation [55]. A recent study reported comparison of cryopreserved and RT stored PC after transfusion in polytraumatic patients (*n* = 25 and *n* = 21, respectively). The data show that despite a twofold lower platelet count after transfusion of cryopreserved compared to RT stored platelets (*p* = 0.02) no differences in hematologic outcomes were found [47]. The measured outcomes were 30-day survival, number of administered blood products, fibrinogen concentrate, tranexamic acid administration and adverse events. The recent CLIP-I trial in Australia [56] was a pilot study to primarily test feasibility and safety of the protocol. As a secondary outcome they reported increased use of fresh frozen plasma and PC in the cryopreservation cohort (*n* = 23) compared to the RT stored cohort (*n* = 18) [56]. Reasons for differences in outcome between the studies of Bohonek et al. and the CLIP-I trial are unknown but could be related to differences in cryopreservation and thawing method.

Assessment of the platelet contribution to transfusion efficacy in non-bleeding thrombocytopenic patients is different than in actively bleeding patients because administration is prophylactic. In such studies (C)CI is most often the primary outcome although some studies measure bleeding time before and after transfusion. Although most studies showed a decreased bleeding time upon transfusion with cryopreserved platelets, this was not the case for all studies [46].

Finally, a specific cohort of patients is those that suffer from platelet refractoriness. It is believed that platelet refractoriness occurs in chronically transfusion dependent patients who raise an immune response to multiple donor antigens. Introduction of leukoreduction decreased the incidence of platelet refractoriness but still up to 10% of acute leukemia patients are affected by refractoriness [27,57]. Currently, transfusion of phenotypically matched PC is the sole solution for patients suffering from platelet refractoriness. In 2016, Gerber et al. showed that platelet counts in these patients can be increased after transfusion of autologous cryopreserved PC contrary to transfusion of regular ABO-matched, RT stored PC. This suggest that transfusion of autologous cryopreserved platelets may offer a (temporary) solution in cases where phenotypically matched PC are not available [27].

#### 2.3.3. Safety

In early studies safety measures focused on the potential toxicity of DMSO. From the 1970s until 1990s, cryopreserved PC were extensively washed to reduce DMSO administered to the study population. Since the no-wash method high (5–6%) DMSO concentrations are quickly reduced to 400–600 mg per PC after reconstitution in plasma or saline [10]. Only minor adverse events have been directly linked to DMSO administration including metallic taste, sulfuric breath odor and nausea [46,58].

Because cryopreserved platelets are procoagulant, concerns for thromboembolic events after transfusion were raised. However, so far, neither thromboembolic events nor any other serious adverse events were reported in trials designed to study the safety of cryopreserved platelets [46,56,58].

## 3. Changes to the GPIbα Receptor

Cryopreservation modifies GPIbα in platelets resulting in two (often) well separated platelet subpopulations in flow cytometry based on CD42b fluorescence intensity. One subpopulation has abnormally low CD42b fluorescence designated GPIbα^low^ and the other has normal CD42b fluorescence, GPIbα^normal^. The GPIbα^low^ platelet subpopulation expresses high PS/PE, responds poorly to agonists measured by PAC-1 binding and has increased factor V binding. On the contrary PS/PE expression is relatively low, PAC-1 binding is high after agonist stimulation and factor V is not as high in the GPIbα^normal^ subpopulation [17,26].

The relative fractions of GPIbα^low^ vs. GPIbα^normal^ platelet subpopulations in PC differed between products prepared by blood institutions in the USA, Australia and Belgium [17]. High GPIbα^low^ content implied low platelet function and vice versa. For instance platelet integrin activation, agglutination and platelet adhesion to collagen were significantly lower in PC with a high GPIbα^low^/GPIbα^normal^ ratio compared to PC with a low ratio [17]. It is not clear why this is or what processes underlie the differences observed between blood institutions.

To investigate the role of GPIbα subpopulations, we inhibited ectodomain shedding during cryopreservation using the matrix metalloproteinase inhibitor marimastat. This hydroxamic acid inhibits A Disintegrin And Metalloproteinase 17 (ADAM17) which is the membrane bound enzyme that catalyzes GPIbα ectodomain shedding [59].

Our data show that marimastat (10 µM) successfully shifted the CD42b fluorescence profile in cryopreserved PC from 52% (±2) in the GPIbα^normal^ subpopulation without inhibitor to 91% (±1) with inhibitor (Figure 2A). Western blotting of PC supernatant confirmed that this was a consequence of shedding inhibition (Figure 2B). Cryopreservation caused a 2.3-fold increase in glycocalicin levels compared to paired samples before freezing. This was completely inhibited in the presence of 10 µM marimastat. Despite complete inhibition of GPIbα shedding, cryopreservation in the presence of marimastat could however not rescue the phenotype of cryopreserved platelets. Agglutination using 1.25 mg/mL ristocetin was 32% and 29% (amplitude) in the absence and presence of 10 µM marimastat, respectively compared to 54% before freezing (Figure 2C). Adhesion rates of cryopreserved platelets to collagen in microfluidic flow chambers were significantly lower compared to paired samples before cryopreservation (*p* = 0.03) and this was irrespective of marimastat addition (*p* = 0.98, *n* = 3) (Figure 2D,E). These data show that GPIbα ectodomain shedding is a consequence of platelet cryopreservation and that its inhibition cannot improve platelet function in agglutination and adhesion experiments. GPIbα expression levels have also been linked to platelet circulation time, but this was not addressed in our study. The effect of shedding inhibition on this parameter thus is not known. It may be interesting to investigate this because Barnard et al. showed that the 2-h posttransfusion recovery of cryopreserved baboon platelets was 48% and mostly attributed to the GPIbα^low^ subpopulation which was cleared from circulation before 1 h after transfusion, while this was not the case for GPIbα^normal^ subpopulation. In addition, circulation time of the GPIbα^normal^ subpopulation was almost 6 days [26]. The Valeri group compared two cryopreservation protocols for human platelets. One protocol caused approximately 20% GPIbα^low^ platelets and the other 50% GPIbα^low^ platelets and corresponding in vivo recoveries were 35–40% and 25–30%, respectively. No differences in circulation time were found [10]. There is currently no direct evidence that the lower recovery is a consequence of the presence of the larger GPIbα^low^ subpopulation but together with the data from Barnard et al. we consider it worthwhile investigating.

## 4. Perspectives

Significant progress has been made in the search for alternative storage methods for platelets but important information is still lacking before cryopreserved platelets can be used in routine blood banks. The no-wash protocol [10] still uses DMSO as a cryoprotectant and patients receiving multiple transfusions may still be exposed to large quantities of this toxic solvent. Alternatives have been investigated but none seem superior to DMSO [60]. Apart from the cryoprotectant, a substantial number of other variables influences the quality of cryopreserved platelets. In addition, the large number of inherent variables that contribute to a successful blood product make comparison between clinical studies challenging in anyway. Consequently, future efforts should also go to the selection of a standard clearly-defined cryopreservation protocol. This may help to design more reliable clinical trials with outcomes that will be more comparable. Until then, safety and efficacy trials will be required in every region where cryopreservation is required. So far, biochemical investigation of signal transduction in cryopreserved platelets is very limited. Such studies may be crucial in unravelling the pathways that are affected. This understanding may help explain the discrepancy between the decreased platelet function in vitro and the encouraging data from clinical trials. It may also help to design novel additive solutions that contain molecules to protect affected pathways. Finally, next to their established role in hemostasis platelets contribute to many other processes including inflammation, (lymph)angiogenesis, cancer and wound healing. Transfusion of (cryopreserved) platelets is underinvestigated in these fields and requires special attention from the research community in the future.

## Figures and Tables

**Figure 1 ijms-21-00935-f001:**
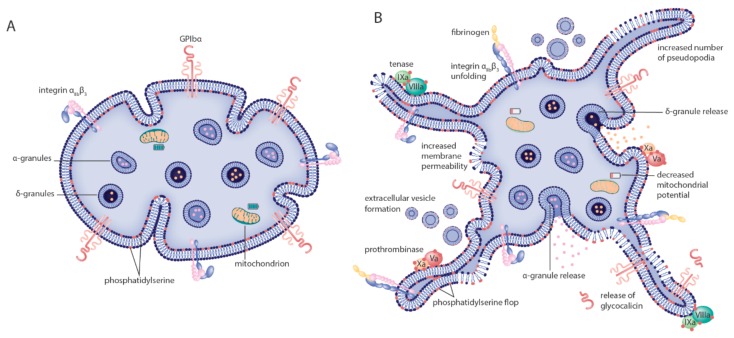
Morphologic and biochemical changes to platelets after cryopreservation. The model indicates known morphologic and biochemical changes to platelets going from (**A**) resting, healthy cells before cryopreservation to (**B**) altered phenotype after cryopreservation. Morphologic alterations induced by cryopreservation include a significant shape change from discoid to spherical platelets and increased numbers of platelet pseudopodia. Main changes to the cytoplasmatic membrane are increased permeability, extracellular vesicle formation and phosphatidylserine flop from the inner to the outer part of the bilayer. The latter two catalyze fibrin formation by providing a binding surface for the tenase and prothrombinase coagulation factor complexes. Many platelet surface receptors are expressed differently after cryopreservation. P-selectin expression is increased, GPIbα expression is decreased and the integrin α_IIb_β_3_ is (in part) activated each marking events of granule content release, receptor ectodomain shedding and fibrinogen binding, respectively. Finally, metabolic changes are detected by defective mitochondrial function.

**Figure 2 ijms-21-00935-f002:**
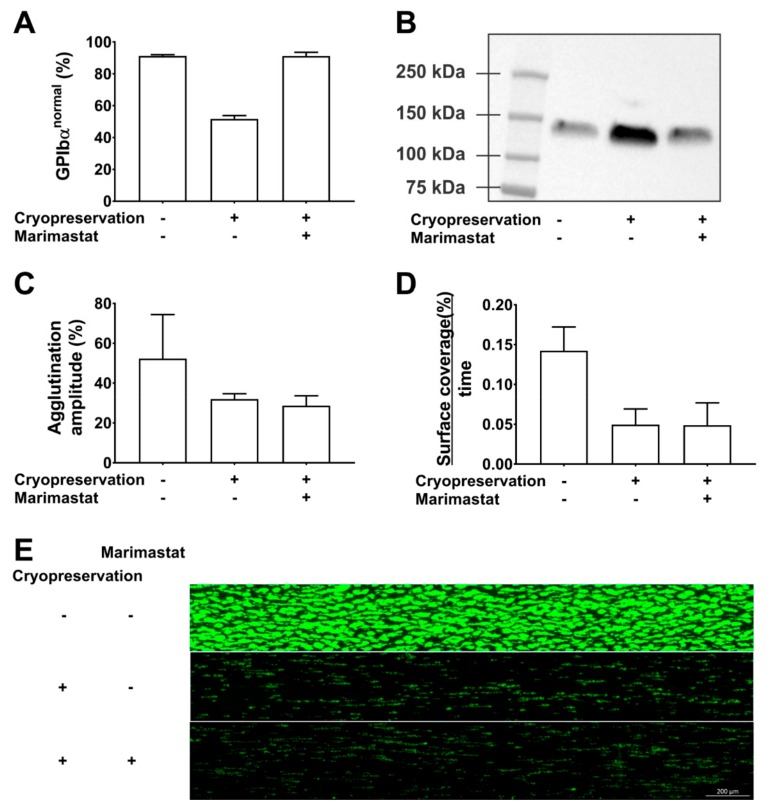
Inhibition of GPIbα shedding during cryopreservation. The effect of cryopreservation in the presence or absence of 10µM marimastat was investigated. Comparison was with paired samples before cryopreservation. The methods used have been described in detail before [17,37,40]. (**A**) The percentage of GPIbα^normal^ platelets was determined by CD42b measurements in flow cytometry. (**B**) Glycocalicin levels were determined by western blotting of platelet-free supernatant. (**C**) The agglutination amplitude of the platelets was measured in light transmission aggregometry using 1.25 mg/mL ristocetin. (**D**) The platelet surface coverage (%) as a function of time was determined by video microscopy of microfluidic flow chambers coated with collagen. Reconstituted whole blood spiked with 250,000 fluorescently labeled platelets per µL was perfused at a wall shear rate of 1000 s^−1^. (**E**) Representative endpoint image of fluorescently labeled adherent platelets after 6 min of perfusion.

**Table 1 ijms-21-00935-t001:** Summary of morphologic and biochemical characteristics and their impact on platelet function of RT stored or cryopreserved platelets.

	RT Stored Platelets	Cryopreserved Platelets
***General***		
Storage conditions	22 °C with constant agitation	−80 °C without agitation
Shelf-life	5–7 days	At least 2 years
***Platelet Characteristics***		
Morphology	Disc	Sphere
Cytoplasmic membrane	Membrane asymmetry	PS/PE expression Loss of membrane integrity
Degranulation	Minimal	Increased
Metabolism	Normal	Loss of mitochondrial membrane potential Decreased hypotonic shock response Accelerated glycolysis
GPIbα shedding	Minimal	Increased
EV formation	Minimal	Increased
***Platelet Function***		
PAC-1 binding upon stimulation with agonists	Normal	Decreased
Aggregation response upon stimulation with agonists	Normal	Decreased
Adhesion rate to ECM under flow	Normal	Decreased
Coagulation rate under flow	Normal	Increased
ROTEM		
Clotting time	Normal	Shortened
Clot firmness	Normal	Slightly decreased
TGA		
Peak thrombin	Normal	Increased
Lag time	Normal	Shortened

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
