# Peer review of "Platelet Biochemistry and Morphology after Cryopreservation"

_ijms, 2020, doi:10.3390/ijms21030935_

Round 1

Reviewer 1 Report

This review provides a comprehensive and up to date overview of current research on the biochemical and molecular changes in platelets following cryopreservation. I only have a few comments as follows

L130: Phosphatidylethanolamine is a natural rather than an anionic phospholipid.

L225: The full name of TRAP-6 should be “thrombin receptor agonist peptide 6”.

L303: please indicate the full name of PAR.

Author Response

response: Thanks to the reviewers for carefully examining our review. All concerns and remarks were answered and the manuscript was amended where requested. Changes are highlighted in bold. We think the review process has significantly improved the manuscript and look forward to further decisions.

L130: Phosphatidylethanolamine is a natural rather than an anionic phospholipid.

response: The reviewer is correct, PE is a neutral phospholipid. The sentence was adapted with removal of the statement about the charge of the phospholipids.

L225: The full name of TRAP-6 should be “thrombin receptor agonist peptide 6”.

response: The abbreviation of TRAP-6 was adapted according to the suggestion by the reviewer in both the text and in the abbreviation list.

L303: please indicate the full name of PAR.

response:The full name of PAR (protease-activated receptor) was included in the text and was added to the abbreviation list.

Reviewer 2 Report

The manuscript is a review of cryopreservation as a platelet storage method for use in platelet transfusion therapy and the impact of cryopreservation on platelet biochemistry.

This is a generally well-written manuscript by authors who have done work in the field. Not many other reviews have been published on this topic, and there has been some recent progress in the field--so a review is timely and of potential interest to practitioners of transfusion medicine and investigators in transfusion science. There are a few instances where phrasing is confusing and word usage is incorrect, which should be corrected. A couple statements are questionably supported by the data and should be rephrased or deleted, but overall the review appears to accurately reflect the current state of the field. The manuscript could benefit from the inclusion of an additional table. Specific comments are as follows:

1. Lines 11-12: This first sentence of the abstract is awkwardly phrased. The authors should consider rephrasing to something like, "Platelet cryopreservation has been investigated for several decades as an alternative to room temperature storage of platelet concentrates."  

2. Lines 98-103: A disc-to-sphere shape change actually is expected to increase platelet volume, because a sphere is the shape with the largest volume for a given surface area. [Also, the volume of a sphere is (4/3)(pi)(r3), whereas a disc has volume (pi)(r2)(h). In a disc, h is less than r (otherwise it would be a cylinder), therefore the volume of a sphere is greater than the volume of a disc of the same radius.] Whether automated hematology analyzers overestimate the volume increase associated with a disc-to-sphere shape change is another issue. Unless the authors can reference empirical evidence of no increase in volume after platelet shape change, they should rewrite this paragraph. The issue of volume change might best be omitted altogether; it isn't an important point for this review.    

3. Lines 111-113: The statement that "it is probably inappropriate to directly interpret the loss of swirl in cryopreserved platelets as a loss of quality similar to RT stored platelets" is unsupported. In both types of platelet product the loss of swirl could represent the disc-to-sphere shape change that occurs with platelet activation. Also, there are highly similar biochemical changes associated with disc-to-sphere shape changes in cryopreserved platelets and room temperature stored platelets. Thus, loss of swirl could in fact represent the same phenomenon in both products. I recommend deleting lines 111-113.

4. Lines 375-376: I recommend rephrasing the sentence that begins with, "Differences in outcomes..." to something like, "Reasons for differences between the studies of Bohonek et al and the CLIP-I trial are unknown, but could be related to differences in cryopreservation and thawing methods." The original phrasing implies more strongly that the differences in methods could account for the different outcomes, but that is  speculation and the reason is unknown.

5. Line 384: The authors should provide a reference for the statement that the population of patients with platelet refractoriness "is increasing in hospitals worldwide."

6. Lines 432-434:The sentence beginning with "Because changes in..." doesn't make sense as written. It may be missing a word.

7. Lines 459-462: The phrasing of the two sentences beginning with "The search for..." is confusing and somewhat disjointed. This should be rephrased. The authors might consider something like, "Efforts to optimize the quality of cryopreserved platelets are complicated by the large number of variables that contribute to a successful product. The high number of variables also makes comparison between clinical studies challenging."

8. Lines 465: The clause, "where cryopreservation is aspired" is an incorrect use of the word "aspired." Consider changing to, "where different cryopreservation protocols are used."

9. The review would be improved with the addition of a new table that summarizes the differences between cryopreserved platelets and room temperature stored platelets, such as differences in storage conditions and shelf life,  characteristics (morphologic, physical, biochemical), biological activity, recovery and survival time, and expected clinical applications.

10. Minor changes:

Line 46: "e.g." is used incorrectly. Consider just deleting "e.g."

Line 55:"Implied" isn't the correct word here. Please rephrase.

Line 57: Change "Research to..."  to "Research into..."

Line 99: Eliminate parentheses around the word "Electron."

Line 463: Change "clear-defined" to "clearly-defined."

Author Response

response: Thanks to the reviewers for carefully examining our review. All concerns and remarks were answered and the manuscript was amended where requested. Changes are highlighted in bold. We think the review process has significantly improved the manuscript and look forward to further decisions.

Lines 11-12: This first sentence of the abstract is awkwardly phrased. The authors should consider rephrasing to something like, "Platelet cryopreservation has been investigated for several decades as an alternative to room temperature storage of platelet concentrates."  

response:Thanks for the suggestion, the sentence was adapted to make the statement more clear.

Lines 98-103: A disc-to-sphere shape change actually is expected to increase platelet volume, because a sphere is the shape with the largest volume for a given surface area. [Also, the volume of a sphere is (4/3)(pi)(r3), whereas a disc has volume (pi)(r2)(h). In a disc, h is less than r (otherwise it would be a cylinder), therefore the volume of a sphere is greater than the volume of a disc of the same radius.] Whether automated hematology analyzers overestimate the volume increase associated with a disc-to-sphere shape change is another issue. Unless the authors can reference empirical evidence of no increase in volume after platelet shape change, they should rewrite this paragraph. The issue of volume change might best be omitted altogether; it isn't an important point for this review.

response: The paragraph was adapted and now includes the discrepancy between increase in platelet volume and unchanged platelet sized when determined by hematology analyzers and on EM images, respectively. We excluded the speculative statement whether automated hematology analyzers overestimate the volume of platelets after shape change.

Lines 111-113: The statement that "it is probably inappropriate to directly interpret the loss of swirl in cryopreserved platelets as a loss of quality similar to RT stored platelets" is unsupported. In both types of platelet product the loss of swirl could represent the disc-to-sphere shape change that occurs with platelet activation. Also, there are highly similar biochemical changes associated with disc-to-sphere shape changes in cryopreserved platelets and room temperature stored platelets. Thus, loss of swirl could in fact represent the same phenomenon in both products. I recommend deleting lines 111-113.

response: Lines 111 to 113 were deleted from the manuscript as recommended.

Lines 375-376: I recommend rephrasing the sentence that begins with, "Differences in outcomes..." to something like, "Reasons for differences between the studies of Bohonek et al and the CLIP-I trial are unknown, but could be related to differences in cryopreservation and thawing methods." The original phrasing implies more strongly that the differences in methods could account for the different outcomes, but that is  speculation and the reason is unknown.

response:The sentence was adapted to ‘Reasons for differences in outcome between the studies of Bohonek et al and the CLIP-I trial are unknown but could be related to differences in cryopreservation and thawing method.‘. (Lines 374 – 376)

Line 384: The authors should provide a reference for the statement that the population of patients with platelet refractoriness "is increasing in hospitals worldwide."

response:Two references were included and the text was modified to ‘Introduction of leukoreduction decreased the incidence of platelet refractoriness but still up to 10% of acute leukemia patients are affected by refractoriness’. (Lines 384 – 385)

Lines 432-434:The sentence beginning with "Because changes in..." doesn't make sense as written. It may be missing a word.

response:The sentence was adapted to ‘GPIbα expression levels have also been linked to platelet circulation time, but this was not addressed in our study. The effect of shedding inhibition on this parameter thus is not known.’. (Lines 435 – 437)

Lines 459-462: The phrasing of the two sentences beginning with "The search for..." is confusing and somewhat disjointed. This should be rephrased. The authors might consider something like, "Efforts to optimize the quality of cryopreserved platelets are complicated by the large number of variables that contribute to a successful product. The high number of variables also makes comparison between clinical studies challenging."

response:The sentence was adapted to ‘Apart from the cryoprotectant, a substantial number of other variables influences the quality of cryopreserved platelets. In addition, the large number of inherent variables that contribute to a successful blood product make comparison between clinical studies challenging in anyway.’. (Lines 452 – 454)

Lines 465: The clause, "where cryopreservation is aspired" is an incorrect use of the word "aspired." Consider changing to, "where different cryopreservation protocols are used."

response:The sentence is adapted to ‘Until then, safety and efficacy trials will be required in every region where cryopreservation is required.’ (Line 457)

The review would be improved with the addition of a new table that summarizes the differences between cryopreserved platelets and room temperature stored platelets, such as differences in storage conditions and shelf life,  characteristics (morphologic, physical, biochemical), biological activity, recovery and survival time, and expected clinical applications.

response:A table was included in the manuscript to summarize the main biochemical and morphological changes of platelets after cryopreservation together with the impact on determinants of platelet function. We did however not include any clinical parameters in this table as this is beyond the scope of this narrative review. So far, clinical data are mainly observational or are conducted on small patient cohorts with diverse clinical indications thus summarizing this in one table may seem biased.

Minor changes:

response: All minor changes were adapted according to the suggestions of the reviewer.

Line 46: "e.g." is used incorrectly. Consider just deleting "e.g."

Line 55:"Implied" isn't the correct word here. Please rephrase.

Line 57: Change "Research to..."  to "Research into..."

Line 99: Eliminate parentheses around the word "Electron."

Line 463: Change "clear-defined" to "clearly-defined."